# The determinants of genetic diversity in butterflies

Alexander Mackintosh[1], Dominik R. Laetsch [1], Alexander Hayward[2], Brian Charlesworth[1], Martin Waterfall[1], Roger Vila[3] & Konrad Lohse [1]

Under the neutral theory, genetic diversity is expected to increase with population size. While comparative analyses have consistently failed to find strong relationships between census population size and genetic diversity, a recent study across animals identified a strong correlation between propagule size and genetic diversity, suggesting that r-strategists that produce many small offspring, have greater long-term population sizes. Here we compare genome-wide genetic diversity across 38 species of European butterflies (Papilionoidea), a group that shows little variation in reproductive strategy. We show that genetic diversity across butterflies varies over an order of magnitude and that this variation cannot be explained by differences in current abundance, propagule size, host or geographic range. Instead, neutral genetic diversity is negatively correlated with body size and positively with the length of the genetic map. This suggests that genetic diversity is determined both by differences in long-term population size and the effect of selection on linked sites.

[1] Institute of Evolutionary Biology, University of Edinburgh, Edinburgh EH9 3FL, UK. [2] Centre for Ecology and Conservation, University of Exeter, Penryn Campus, Cornwall TR10 9FE, UK. [3] Institut de Biologia Evolutiva (CSIC Universitat Pompeu Fabra), Passeig Marítim de la Barceloneta 37, ESP-08003 Barcelona, Spain. Correspondence and requests for materials should be addressed to A.M. (email: alexmackintosh44@gmail.com) or to K.L. (email: konrad.lohse@ed.ac.uk)

The genetic diversity segregating within a species is a central quantity; it determines its evolutionary potential, and is, in turn, the outcome of its selective and demographic past. Under the neutral theory[1] genetic diversity is expected to be proportional to the product of the effective population size, $N_e$, and the per-generation mutation rate, $\mu$[2] (provided that $N_e\mu$ is sufficiently small that the infinite sites mutation model is applicable[3]). Given that census population size varies widely across the tree of life, much of the variation in genetic diversity between species should be due to differences in census size. However, correlates of census size, such as geographic range, have repeatedly been found to be poor predictors of genetic diversity[4–7]. In addition, genetic diversity seems to vary remarkably little overall, given the wide range of population sizes seen in nature. Although the extremely narrow ranges of genetic diversity reported by early comparative studies based on allozymes[6] are partly explained by balancing selection[8], diversity at nearly neutral sites is also restricted to a narrow range of two orders of magnitude[4] (with some notable exceptions[5,9]). While the fact that there are only four alternative states for a nucleotide site implies a hard upper bound on the possible level of nucleotide site diversity (of 0.75 assuming no mutational bias)[10], the levels of neutral genetic diversity seen in natural populations remain far below this.

The observation that genetic diversity does not correlate with measures of census size, known as Lewontin's paradox, has intrigued evolutionary biologists for nearly half a century. Proposed solutions to the paradox are generally of two types: the first proposes that there may be a negative relationship between $N_e$ and $\mu$[11], and the second seeks reasons why $N_e$ shows such little variation between species[12]. Given the lack of firm evidence for large differences in mutation rate among species with different levels of variability, recent comparative studies have focused on identifying factors that determine long-term $N_e$ and hence genetic diversity[13].

One explanation for the narrow range of genetic diversity observed in nature is that natural selection continuously removes neutral diversity linked to either beneficial[12] or deleterious variants[14,15]. Because the efficacy of selection depends on the product $N_e s$, selection is expected to be more efficient and therefore remove more neutral linked sites in species with large $N_e$. Recently, Corbett-Detig et al.[16] have shown that the proportional reduction of neutral diversity due to selection at linked sites does indeed correlate with measures of census size such as geographic range and (negatively) with body size. While Corbett-Detig et al.[16] argue that this can explain "… why neutral diversity does not scale as expected with census size", a reanalysis of their data[17] concluded that the effect of selection on linked neutral diversity is too small to provide a general explanation for the narrow range of genetic diversity seen in nature.

An alternative (but not mutually exclusive) explanation is that variation in genetic diversity is constrained by fluctuations in long-term population size. This would imply that genetic diversity should correlate with life-history traits that affect a species susceptibility to environmental fluctuations. Romiguier et al.[5] and Chen et al.[18] have uncovered a striking negative correlation between propagule size and genetic diversity across the animal kingdom: species that are short-lived and invest little into many offspring (r-strategists) tend to have higher genetic diversity than long-lived species with few offspring and large parental investment (K-strategists). They argue that K-strategists may be able to avoid extinction at low population sizes, while r-strategists require much larger populations to buffer against environmental fluctuations. While it is striking that propagule size explains 70% of variation in genetic diversity across animal phyla, this finding is unlikely to explain variation in genetic diversity in taxa with similar life-history strategies. However, if robustness to fluctuations in population size is the ultimate determinant of genetic diversity—as Romiguier et al.[5] argue—one would expect other life-history traits to correlate with genetic diversity. In particular, more specialized species may be able to avoid extinction in spite of small census sizes and thus have reduced long-term $N_e$. Alternatively, if the efficacy of selection determines intraspecific genetic diversity then variation between species may be best explained by differences in recombination rate and the density of selective targets in the genome.

Here we address these questions using butterflies (Papilionoidea) as a model system. Papilionoidea share a common ancestor ~119 million years ago (MYA)[19], and are characterised as r-strategists given their short life span and high fecundity[20], with little variation in reproductive strategy. Butterflies, in particular European species on which we focus, are arguably the best studied group of insects. Thanks to centuries of study by scientists and amateur naturalists together with numerous recording schemes, butterfly taxonomy, geographic ranges and life-histories are known in great detail. This offers a unique opportunity not only to revisit potential correlates of genetic diversity that have proven difficult to quantify across large taxonomic scales (e.g. geographic range and abundance), but also test the effects of ecological traits. In particular, while niche breadth is difficult to quantify across distantly related taxa and has not so far been considered in comparative analyses of genetic diversity, accurate data for the number of larval host plants (LHP) exist for European butterflies.

We estimated genetic diversity from de novo transcriptome data for 38 butterfly species (sampling two individuals from each, Supplementary Data 1). For simplicity, the estimation of synonymous diversity was restricted to fourfold degenerate sites ($\pi_{4D}$) and non-synonymous diversity was estimated at zero-fold degenerate sites ($\pi_{0D}$), i.e. sites where any nucleotide change leads to an amino acid difference. While synonymous sites are subject to codon usage bias[21,22], they are not directly affected by other forms of selection. Although selection intensities on codon usage and biased gene conversion at synonymous sites have been little studied in Lepidoptera[23], recent population genomic studies of Drosophila species suggest that the product of $N_e$ and selection coefficients for such sites is generally of the order of 1 or less, so that they can be treated as nearly neutral[24]. Our rationale for modelling $\pi_{4D}$ and $\pi_{0D}$ jointly was to better understand the nature of the underlying forces at the population level: theory predicts that any correlate of neutral genetic diversity ($\pi_{4D}$) that increases $N_e$ should correlate less strongly with diversity at non-synonymous sites ($\pi_{0D}$)[25]. This is because any increase in diversity due to reduced genetic drift is counteracted by the removal of diversity due to more efficient selection. In contrast, any trait that affects non-synonymous genetic diversity ($\pi_{0D}$) via the absolute strength of selection $s$ should be more strongly correlated with diversity at non-synonymous sites ($\pi_{0D}$) than synonymous sites ($\pi_{4D}$), which are only indirectly affected.

We investigate the relation between average nucleotide site diversity[26] and five ecological traits: census size (estimated as the product of abundance and geographic range), body size, voltinism, egg volume (relative to body size) and LHP breadth (Source Data) using a generalized linear mixed model. In addition, we test whether genome size and recombination rate affect genetic diversity. In the absence of detailed recombination maps, we use the number of chromosomes as a proxy for the length of the genetic map. This assumes an average map length of 50 cM per chromosome in male meiosis, and takes into account the lack of crossing over in female meiosis of Lepidoptera. This assumption is supported by a linkage map for the butterfly Heliconius melpomene[27] as well as the silkmoth Bombyx mori[28]. We find that neutral genetic diversity across butterflies varies over an order of

magnitude. Perhaps surprisingly, this variation cannot be explained by differences in current abundance, propagule size, host or geographic range. Instead, genetic diversity correlates negatively with body size and positively with the length of the genetic map. This suggests that levels of genetic diversity are determined both by long-term population size and the effect of selection on linked sites.

## Results

**Neutral diversity varies over an order of magnitude.** Genetic diversity was estimated for 38 species of European butterfly from five families: Papilionidae, Hesperiidae, Pieridae, Lycaenidae and Nymphalidae (Fig. 1). For 33 species, we generated and de novo assembled short read RNA-seq data for two individuals; for five species raw RNA-seq reads were downloaded from a previous study[5] (Supplementary Data 3). Variants in each species were called by mapping reads back to reference transcriptomes. Only transcripts present in a set of 1277 single-copy orthologues (SCOs), which we identified from the 33 transcriptomes with high completeness (BUSCO scores 96.3–98.4%, Supplementary Fig. 1, Supplementary Data 3), contributed to estimates of genetic diversity. Neutral genetic diversity as measured by $\pi_{4D}$ varies over an order of magnitude across this set of butterfly species: from 0.0044 in *Pieris brassicae* (the cabbage white) to 0.0428 in *Spialia sertorius* (the red-underwinged skipper) (Fig. 1). However, the mean ($\pi_{4D} = 0.0175$) is typical of insects[4,16,18]. Assuming neutrality and a per-site per-generation spontaneous mutation rate of $\mu = 2.9 \times 10^{-9}$ [29], this corresponds to $N_e$ on the order of $10^5$–$10^6$ individuals, a much lower range than that reported for distantly related animal taxa[5,18]. While Romiguier et al.[5]—sampling across the entire animal kingdom—found that species in the same taxonomic family have similar genetic diversity, we observed no significant family effect in butterflies (ANOVA, $F_{4,33} = 1.841$, $p = 0.144$). More generally, phylogeny was a poor predictor of neutral genetic diversity in butterflies ($n = 38$, Pagel's $\lambda = 7.4 \times 10^{-5}$, $p = 1$, assuming that $\pi_{4D}$ evolves in a random walk along the phylogeny, Fig. 1).

**Non-synonymous diversity and the efficacy of selection.** Since directional selection will purge (or fix) mutations at non-synonymous sites[30], we expect diversity at these sites to be greatly reduced compared with synonymous sites. Within this set of butterfly species, $\pi_{0D}$ and $\pi_{4D}$ typically differ by an order of magnitude as commonly found in insects. Under the nearly neutral theory[31] and assuming a gamma distribution for the distribution of mutational effects on fitness (DFE), the slope of the negative linear relationship between $ln(\pi_{0D}/\pi_{4D})$ and $ln(\pi_{4D})$ is equal to the shape parameter, $\beta$[25]. The slope we estimate in butterflies (Supplementary Fig. 2) implies that there is a substantial fraction of weakly deleterious mutations ($\beta = 0.45$, 95% CI = 0.37–0.53). This is higher than the estimates for *Heliconius* butterflies (0.08–0.28)[18], but compatible with previous estimates of the DFE for *Drosophila* based on the site frequency spectrum[32].

In contrast to previous comparative studies, we restricted our analysis to SCOs shared by all species. While this eliminates noise when comparing genetic diversity between species, it invariably introduces a bias towards highly expressed and well conserved genes. Thus, our estimates of $\pi_{0D}$ are almost certainly underestimates of genome-wide non-synonymous diversity (Supplementary Fig. 3). In contrast, we find that $\pi_{4D}$ values estimated from SCOs are only slightly lower than estimates based on all genes (Supplementary Fig. 3), suggesting that codon usage bias has little effect on our estimates of putatively neutral diversity.

**Nuclear and mitochondrial diversity are uncorrelated.** Mitochondrial (mt) genes are an easily accessible source of variation

data and have been extensively used to infer the phylogeographic history of species and populations[33,34]. However, it is becoming increasingly clear that variation in mt diversity largely reflects selective processes and variation in mt mutation rates[35] rather than the rate of genetic drift[36,37]. In groups with Z/W sex determination, such as butterflies, mt diversity may be additionally reduced by selection acting on the W chromosome (which is co-inherited with the mitochondrion)[38]. Several comparative studies have shown that mt diversity is uncorrelated with measures of abundance and nuclear diversity[36,37,39]. We find that across European butterflies, mt diversity at the *COI* barcode locus is only very weakly (and not significantly) correlated with both $\pi_{4D}$ (Pearson's correlation, $d.f. = 36$ $r = 0.149$, $p = 0.371$) and $\pi_{0D}$ ($r = 0.257$, $p = 0.119$, Supplementary Fig. 4).

**No effect of abundance or life history on genetic diversity.** Estimates of census population size are uncorrelated with both $\pi_{0D}$ and $\pi_{4D}$ (Supplementary Table 1). This suggests that present day ranges and abundance have little to do with long-term $N_e$ in butterflies and mirrors the findings of Romiguier et al.[5] across the animal kingdom. However, unlike Romiguier et al.[5] and Chen et al.[18], who have found a strong negative correlation between propagule size and neutral genetic across species, we find no significant effect of relative egg size (egg volume/body size) on $\pi_{4D}$ (Supplementary Table 1). Similarly, voltinism is not significantly correlated with $\pi_{4D}$ ($p = 0.159$, Supplementary Table 1), however, the trend towards polyvoltine taxa having greater $\pi_{4D}$ is at least consistent with the idea that r-strategists have larger long-term $N_e$[5]. We also find that larval host plant (LHP) breadth has no significant effect on $\pi_{4D}$ or $\pi_{0D}$ (Supplementary Table 1). This is true regardless of whether we classify species as monophagous if all LHPs are within one family (and polyphagous otherwise) or, instead, consider the number of LHP species as a predictor (Supplementary Fig. 5).

Only one trait, body size, is significantly and negatively correlated with $\pi_{4D}$ ($p = 0.003$, Table 1, Fig. 2a): smaller butterfly species tend to have higher genetic diversity. This correlation is significant after Bonferroni correction for multiple testing. As predicted for correlates of long-term $N_e$, the effect is weaker for $\pi_{0D}$ (Table 1) than $\pi_{4D}$. We can express the effects of body size on $ln(\pi_{4D})$ and $ln(\pi_{0D})$ in terms of $ln(\pi_{0D}/\pi_{4D})$. This ratio is weakly and positively correlated with body size (posterior mean slope = 0.120, $p = 0.049$), suggesting that selection is more efficient in smaller species.

**Chromosome number correlates with genetic diversity.** While $\pi_{4D}$ correlates positively and significantly with chromosome number (posterior mean slope = 0.279, $p = 0.004$, Table 1, Fig. 2b), it is not significantly correlated with genome size, i.e. the physical length of the genome (estimated using flow cytometry, see Methods section) (Supplementary Table 1). Assuming that the number of genes in the genome (and other potential targets of selection) is more or less constant and independent of genome size, population genetic theory predicts the aggregate effect of selection on linked neutral diversity to be largely determined by the map length of a chromosome, for a given set of selection and mutation parameters[15,40] (see Discussion section).

Although (unsurprisingly) the effect of chromosome number we find depends disproportionately on the two species with the fewest chromosomes (*Pieris brassicae*, $n_c = 15$, and *Melanargia ines* (the Spanish marbled white), $n_c = 13$, Fig. 2b), removing both species still gives a positive (albeit non-significant) relation between genetic diversity and chromosome number (posterior mean slope = 0.181, $p = 0.117$).

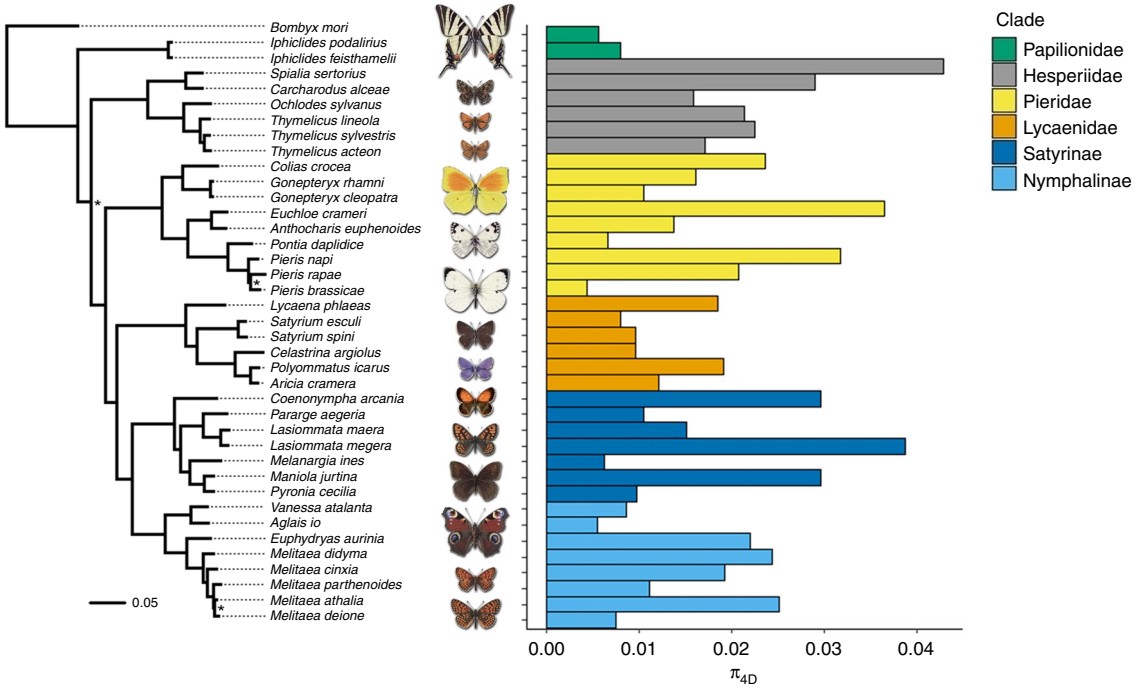

**Fig. 1** Neutral genetic diversity ($\pi_{4D}$) across European butterfly species. The phylogeny is based on 218 single-copy orthologues and rooted with the silkmoth *Bombyx mori* as an outgroup. All nodes have 100% bootstrap support unless marked with an asterisk (70–99%). The barplot on the right shows genome-wide estimates of $\pi_{4D}$ for 38 focal species sampled from the six major groups of Papilionoidea present in Europe. The phylogeny explains very little of the variation in $\pi_{4D}$ in butterflies. Source data are provided as a Source Data file

**Table 1 Correlates of genetic diversity inferred under a minimal model**

| Predictor | Response | Posterior mean slope[a] | 95% CI | $p_{MCMC}$[b] |
|---|---|---|---|---|
| Body size | $\pi_{4D}$ | −0.321 | −0.518, −0.114 | 0.003 |
| Body size | $\pi_{0D}$ | −0.201 | −0.330, −0.062 | 0.004 |
| Chrom. number | $\pi_{4D}$ | 0.279 | 0.105, 0.475 | 0.004 |
| Chrom. number | $\pi_{0D}$ | 0.149 | 0.023, 0.266 | 0.017 |

[a]Posterior mean estimates of the slope of linear correlates of genetic diversity
[b]Twice the probability that the posterior mean slope estimate is >0 or <0

**Pleistocene bottlenecks and demography.** Genetic diversity in many European taxa has been shaped by the cycles of isolation into, and range expansion out of, glacial refugia during the Pleistocene[33,34,41]. While we have sought to minimize the effects of Pleistocene history by focusing sampling on a single Pleistocene refugium, Iberia, our inferences could be confounded in at least two ways: Firstly, rather than being solely driven by long-term $N_e$, variation in genetic diversity in Iberia may be affected by gene flow from other refugia[42] or even species[43]. Secondly, even if Iberian populations are little affected by admixture, they may have undergone drastic (and potentially different) changes in $N_e$ in response to past climatic events. Population bottlenecks affect $\pi$, but correspond to a sudden burst in coalescence rather than a change in its long-term rate[44]. Population bottlenecks would also affect our interpretation of $\pi_{0D}/\pi_{4D}$ as a measure of the efficacy of selection. Since $\pi_{0D}$ recovers more quickly than $\pi_{4D}$ after a bottleneck[45], one would expect taxa that have undergone recent changes in $N_e$ to fall above the line of best fit in the relationship between $ln(\pi_{4D})$ and $ln(\pi_0/\pi_{4D})$ (Supplementary Fig. 2).

While modelling demography from our transcriptome data is challenging, the distribution of heterozygous sites in a single diploid individual contains some information about past demography. In particular, an extreme bottleneck or a history of rapid population growth lead to strongly correlated pairwise coalescence times. Considering a fixed length of sequence, we expect the number of heterozygous sites $S$ to be Poisson distributed, whereas intermediate bottlenecks result in multi-modal distribution of $S$ with an increased variance relative to a constant sized population[46]. However, the majority of species show a unimodal, long tailed distribution of $S$, more akin to that expected for a population of constant $N_e$ than the limiting case of an extremely bottlenecked (or rapidly growing) population. In fact, only seven species have a higher variance in $S$ than expected for a population of constant size (Supplementary Fig. 6).

**Robustness to population structure.** The relationship between genetic diversity and population size predicted by the neutral theory assumes a randomly mating population at mutation-drift equilibrium. Since population structure is ubiquitous, an obvious question is to what extent our findings are confounded by differences in population structure across species. For example, the correlation between body size and diversity may simply be a consequence of the reduced dispersal ability of smaller species. If

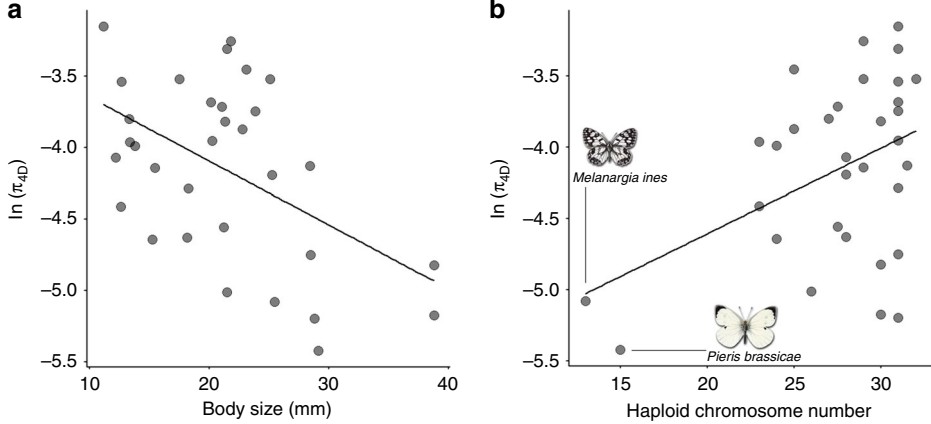

**Fig. 2** The correlates of genetic diversity in butterflies. Neutral genetic diversity $\pi_{4D}$ (**a**) is negatively correlated with body size and positively with the number of chromosomes (**b**). Source data are provided as a Source Data file

this were the case, we would also expect genetic differentiation to correlate with body size. However, we find no evidence for this: differentiation between individuals sampled >50 km apart is low overall (median $F_{IT} = 0.012$) and uncorrelated with body size (Pearson's correlation, $p = 0.684$) (Supplementary Fig. 7). Furthermore, the effect of body size on genetic diversity remains essentially unchanged if we estimate $\pi_{4D}$ and $\pi_{0D}$ within rather than between individuals. Increased population structure in smaller species, therefore, cannot explain the negative relationship between genetic diversity and body size.

Our dataset does include a handful of species with notably high $F_{IT}$ within Iberia, such as *Euphydryas aurinia* (the marsh fritillary) and *Coenonympha arcania* (the pearly heath) ($F_{IT} = 0.259$ and $0.115$, respectively). Interestingly, both species fall above the line of best fit in Supplementary Fig. 2, suggesting that selection is less efficient globally (i.e. $\pi_{0D}/\pi_{4D}$ is higher) in these species. The presence of different locally adapted subspecies or populations could further increase $\pi_{0D}/\pi_{4D}$. For both species, several ecotypes/subspecies exist in the Iberian peninsula, but their exact distribution and status is uncertain. In contrast, the migratory species *Vanessa atalanta* (red admiral) is an outlier in the opposite direction and has lower diversity at non-synonymous sites ($\pi_{0D}$) than expected given its neutral diversity ($\pi_{4D}$) (Supplementary Fig. 2).

## Discussion
We show that neutral genetic diversity in European butterflies varies over an order of magnitude, and that this variation is neither significantly correlated with current abundance nor key life-history traits. In particular, we find little support for the idea that generalist species have larger long-term $N_e$ and hence greater levels of genetic diversity than specialists. We also do not find any significant relationship between propagule size or longevity and neutral genetic diversity. This contrasts with previous comparative studies across larger taxonomic scales[5,18], which have found that (absolute) propagule size is more strongly correlated with genetic diversity than body size and explains 70% of the variation in genetic diversity across animals[5]. We find the opposite pattern in butterflies, that is, $\pi_{4D}$ is much more strongly correlated with body size than (absolute) propagule size and there is no significant relation between relative propagule size and neutral diversity. This suggests that the correlates of genetic diversity are strongly dependent on phylogenetic scale. Clearly, life-history traits that determine differences in genetic diversity over very large taxonomic scales, are likely to be conserved within major

groups of taxa, and so unlikely to explain the still considerable variation in genetic diversity seen within them.

Across European butterflies, we find that body size and chromosome number are the only significant correlates of neutral genetic diversity, and together explain 45% of the variation in genetic diversity. The negative correlation between body size and genetic diversity is consistent with body size limiting population density[47] and therefore long-term $N_e$. This relationship is not exclusive to butterflies, and has been found in mammals[48] and across animals[5] more widely.

As we show below, the positive correlation between chromosome number and neutral genetic diversity is an expected consequence of selection and mirrors the nearly ubiquitous intraspecific correlation between genetic diversity and recombination rate[49,50]. Thus, unlike previous comparative studies which have shown that selection merely constrains variation in genetic diversity[16], our analyses suggest that the effect of selection on linked neutral diversity may explain some of the variation in neutral genetic diversity between taxa that differ in the length of the genetic map.

The lack of any correlation between estimates of census size and $\pi_{4D}$, we find mirror results of previous studies[4–7] and suggests that current abundance does not reflect long-term $N_e$ in butterflies. While the distribution of heterozygosity across the genome suggests that it is unlikely that variation in genetic diversity across butterflies is due to drastic demographic events during the Pleistocene (Supplementary Fig. 6), very recent demographic changes could explain the weak relationship between estimates of census population size and $\pi_{4D}$. In particular, the low genetic diversity of *Pieris brassicae*, a pest species with enormous current population sizes, is compatible with a rapid expansion which may have happened too recently to leave much signal in the data: $Var[S]$ is not particularly low for *P. brassicae* (5.67 compared with the mean among species of 4.85). Interestingly, analysis of RAD-seq data from the closely related species *P. rapae* (the small white) suggests a population expansion ≈20,000 yBP (shortly after the last glacial maximum) followed by divergence into subspecies 1200 yBP when *Brassica* cultivation intensified[51]. It is therefore possible that, by contrast, the ancestral *P. brassicae* population remained small after the glacial maximum and only expanded as recently as ≈1200 yBP.

If variation in carrying capacity shapes genetic diversity in butterflies, it is perhaps surprising that niche breadth, the number of larval host plants (LHPs), is uncorrelated with $\pi_{4D}$. However, given that LHPs vary drastically in geographic range and density, the number of LHPs may be a very crude predictor of a species

long-term census size: a species with a single LHP may have very large populations if its host is widespread. Conversely, a generalist such as *Celastrina argiolus* (the holly blue), one of the most widespread and generalist (>100 LHPs) species in our set, may have low long-term $N_e$ and hence low genetic diversity ($\pi_{4D} = 0.0095$) due to other biotic factors.

There are several potential life-history traits that might have large effects on long-term $N_e$ which we have not considered: in particular, how (in what life-cycle stage) and where species hibernate, the rate of parasitoid attack and the degree of migratory versus sedentary behaviour. Exploring whether these correlate with genetic diversity will require larger sets of taxa.

We have assumed linear relationships between body size and chromosome number and genetic diversity without paying any attention to the causative forces at the population level. To gain some insight into whether the genome-wide effects of background selection (BGS)[14,15] and recurrent selective sweeps[12,52,53] can plausibly explain the observed relationship between diversity and chromosome number, it is helpful to consider analytic predictions for the reduction in neutral diversity due to selection at linked sites. We take as a starting point the expression of Coop (2016) (see ref. [17], Eq. (1)) for the expected genetic diversity given BGS and sweeps occurring homogeneously along the genome. Note that this approximate result assumes independence between selective events and is based on a considerable body of previous population genetics theory[52–54] (Supplementary Note 1):

$$E[\pi] = \frac{\pi_0}{2N_0 J \nu r_c^{-1} + B^{-1}}, \tag{1}$$

where $\pi_0 = 4N_0\mu$ is the genetic diversity in the absence of selection, $B$ is the effect of BGS on diversity, $\nu$ and $r_c$ are the rates of sweeps and recombination per base pair per generation, respectively, in the genomic region under consideration. $J/r_c$ captures the probability of a sweep leading to coalescence at a typical neutral sites. Assuming semi-dominance with selection coefficient $s$ in homozygotes, $J \approx s/[2\ln(2N_e s)]$ (for details, see Supplementary Note 1). We can think of $2N_0 J \nu r_c^{-1}$ as the rate of sweep-induced pairwise coalescence events relative to genetic drift. A simple approximation for the effect of BGS is $B \approx \exp(-U/r_c)$[15], where $U$ is the per base-pair rate of deleterious mutations per diploid genome. Thus both the effects of BGS and positive selection depend on the rate of mutational input relative to recombination. We can scale the rates of deleterious mutations and selective sweeps per genome (rather than per bp). Assuming that the number of selective targets is fixed across species and that there is a linear relationship between recombination rate and map length, the expressions for BGS and positive selection are functions of the number of chromosomes, $n_c$: $\nu/r \approx 4\nu_T/n_c$ and $B \approx \exp(-4U_T/n_c)$, where $\nu_T$ and $U_T$ are the genome-wide rates (on the coalescence timescale) of selective sweeps and new deleterious mutations respectively (Supplementary Note 1).

One immediate conclusion from the above is that, given the large number of chromosomes in butterflies ($13 \leq n_c \leq 31$), BGS can only have a modest effect on neutral diversity: even if we assume a rate of $U_T = 1$ deleterious mutation per genome, the reduction in diversity due to BGS, $B$, only ranges between 0.73 and 0.88 for our dataset. Ignoring the effect of BGS, we have:

$$\frac{E[\pi]}{\pi_0} = (8N_0 J \nu_T n_c^{-1} + 1)^{-1} \tag{2}$$

We can use Eq. (2) to ask how compatible the expected effect of selective sweeps on neutral diversity is with our estimate of the slope of the relation between $\ln(\pi_{4D})$ and $n_c$ (Table 1). In the limit of a high rate of sweeps $\nu_T$, Eq. (2) implies that $\frac{\partial \ln(E[\pi])}{\partial n_c} = n_c^{-1}$; assuming an average of $n_c = 25$ chromosomes, we would expect a maximum slope of 0.04, which is compatible with our empirical

estimates of the slope between $\ln(\pi_{4D})$ and $n_c$ (the estimate in Table 1 corresponds to 0.0620 (95% CI 0.0224, 0.01041) on the untransformed $n_c$).

One can go one step further and use Eq. (2) to estimate the rate of sweeps from the data by minimizing the sum of squared differences between observed and predicted $\pi_{4D}$ across species. If we assume that $N_0$ depends linearly on body size, a spontaneous mutation rate of $\mu = 2.9 \times 10^{-9}$[29] and $J = 10^{-5}$ (which is consistent with estimates of $N_e$ and $s$ in Drosophila[32]), we can co-estimate both the correlation between $N_0$ and body size and $\nu_T$ (Supplementary Software). The best fitting selection regime implies an extremely high rate of sweeps of ($\nu_T \approx 0.133$ per generation). However, this approximate model of the effect of selective sweeps on $\pi$ predicts a much narrower range of $\pi_{4D}$ than is observed (Supplementary Fig. 8). Thus, the above calculation agrees with the analysis of Coop[17], in showing that simple approximations for the effect of selection on neutral diversity cannot on their own explain the variation in genetic diversity among species seen in nature.

We have assumed that chromosome number is simply a proxy for the genetic map length and affects genetic diversity by modulating the effect of selection on linked neutral sites. However, what is cause and effect is far from clear, and chromosome number may itself depend on the efficacy of selection. In particular, a causative relationship between mutation and recombination would be an alternative explanation for the correlation between chromosome number and genetic diversity we find. However, the evidence for this has been very mixed[23,49,55,56]. If recombination was mutagenic, we would expect the two species with strongly reduced chromosome number to have disproportionately low $\pi_{4D}$, i.e. to fall below the line of best fit in Supplementary Fig. 2, which is not the case.

Hill et al.[57] recently found that chromosomes in *Pieris napi* (the green-veined white) are derived from multiple ancestral syntenic blocks, suggesting a series of fission events that was followed by the creation of a novel chromosome organisation through fusions. Given that *P. napi* returned to a karyotype close to the ancestral $n_c = 31$ of butterflies, there may be some selective advantage in organising the genome this way. If this is the case, chromosome rearrangements that produce karyotypes distant from $n_c = 31$ may only be tolerated in populations dominated by drift. While the forces driving karyotype evolution in butterflies are currently not understood, chromosomal fusions accumulate in small populations[58,59] and in selfing plants[60]. Thus, an alternative explanation for the positive correlation between chromosome number and genetic diversity we find is that species with low $N_e$ accumulated mildly deleterious chromosome rearrangements through genetic drift. *Pieris brassicae* ($n_c = 15$) and *Melanargia ines* ($n_c = 13$), which have probably undergone relatively recent chromosomal fusions (given that in both cases relatives in the same genus have higher $n_c$), are consistent with this. As no species in our set has $n_c \gg 31$, we cannot test whether the relationship between genetic diversity and chromosome number is quadratic, and thus consistent with a model where reduced $N_e$ may lead to both increases and decreases in $n_c$. Interestingly, species in the genus *Leptidea*, which have undergone a recent explosion in chromosome number ($n_c$ ranges between 26 and 120[61]), appear to have very low genome-wide diversity ($\pi$ across all site between 0.0011 and 0.0038)[62], consistent with the idea that extreme karyotypes arise during periods of low $N_e$.

Lynch and Conery[63] have put forward analogous arguments for the evolution of genome sizes: genomes may expand in populations with low $N_e$, if selection against transposable element proliferation and intron expansion becomes inefficient. While the large genome size and TE content of *Leptidea* species[62] is consistent with this, we find no support for any relationship between

genome size and neutral diversity across our set of species. Instead, our analyses clearly show that genome size has significant phylogenetic signal across butterflies ($n = 37$, Pagel's $\lambda = 1.000$, $p = 6.1 \times 10^{-7}$) and so must evolve slowly, whereas variation in genetic diversity has little phylogenetic structure (Fig. 1).

While we have only considered a small number of life-history traits and genomic parameters, and have modelled neither the effects of selection nor demography explicitly, it is encouraging that we have identified two simple determinants, which together explain a substantial fraction of the variance in genetic diversity across butterflies. However, a more complete understanding of the processes that shape genetic diversity and how these correlate with life-history will require modelling both the demographic and the selective past explicitly[45,64]. For example, a previous comparative study based on whole-genome data reconstructed the directional histories of divergence and admixture between refugial populations for a different guild of insects[42] and found a trend of refugial population being younger in specialist species. An important next step is to include models of selection and its effects on linked sequences in such inferences. Given sufficiently large samples of taxa, one can then tease apart life-history traits that affect genetic diversity via demographic parameters ($N_e$ in the absence of selection and gene flow between populations) from those that determine the strength of selection itself. Rather than focusing on pairwise $\pi$, the most drastic summary of genetic variation, such inferences will require methods that make use of the rich information contained in genomic data. Furthermore, comparative analyses that are based on whole-genome (rather than transcriptome) data and high quality genome assemblies are required to exploit the extra information about selection and genetic drift that is contained in the genomic context (functional density and direct estimates of the recombination map). Given the detailed knowledge of their taxonomy, ecology, geographic range and their relatively compact genomes, butterflies are one of the best test cases for attempting a reconstruction of the evolutionary processes that result in Lewontin's paradox.

## Methods

**Sampling and sequencing.** Butterflies were hand-netted at various locations across four regions in Iberia (Southern Portugal, Northern Portugal, Catalonia and Asturias, Supplementary Data 1), frozen alive in a liquid nitrogen dry shipper and stored at −80 °C. Two individuals per species were selected for RNA extraction and sequencing. Each species was represented by one female and one male individual whenever possible. Species identities were confirmed by amplifying and sequencing the standard mitochondrial barcode (a 658-bp fragment of COI, primers LepF and LepR[41]) and comparison against a reference database for Iberian butterflies[41] for the following species: *Carcharodus alcae, Coenonympha arcania, Euphydryas aurinia, Melitaea deione, Thymelicus acteon* and *T. sylvestris.*

RNA was extracted using a TRIzol (Ambion) protocol according to the manufacturer's instructions. TruSeq stranded polyA-selected RNA libraries were prepared by Edinburgh Genomics and strand specific 75b paired-end reads were generated on a HiSeq4000 Illumina instrument. Raw reads are deposited at the European Nucleotide Archive (PRJEB31360). RNA-seq datasets for *Melitaea athalia, M. cinxia, M. didyma, M. parthenoides,* and *Thymelicus lineola*—previously analysed in ref. [5]—were retrieved from the European Nucleotide Archive (ENA).

**Read data processing.** Detailed description of the read data processing steps can be found in Supplementary Methods. In brief, quality and adapter trimmed reads were assembled into de novo transcriptomes for both individuals of each species. Protein coding transcripts were identified based on homology information and ORF presence in the CDS. These transcripts were further filtered by read support (read depth ≥10 and MQ ≥1) in both individuals of each species as well as their proteins being single-copy orthologues (SCOs) across all analysed species. The resulting loci were subjected to variant calling. A super-matrix maximum likelihood phylogeny was inferred based on SCOs as described in Supplementary Methods.

**Estimating genetic diversity.** To minimize the confounding effect of population structure (and inbreeding), we calculated $\pi_b$, i.e. the genetic diversity between the

two individuals A and B sampled for each species (analogous to $d_{XY}$):

$$\pi_b = \frac{(n_A + n_B + n_{AB})/2 + n_{fix}}{n_{tot}} \qquad (3)$$

where $n_A$, $n_B$ are the numbers of heterozygous sites unique to A and B, $n_{AB}$ is the count of shared heterozygous sites and $n_{fix}$ is the number of fixed differences. Calculations were carried out separately for fourfold degenerate ($\pi_{4D}$) and zerofold degenerate ($\pi_{0D}$) sites using the script `bob.py` (www.github.com/DRL/mackintosh2019). Estimates of $\pi$ for COI locus of each species were calculated as described in Supplementary Methods.

**Statistical analysis.** Phylogenetic mixed models were constructed using the R package `MCMCglmm`[65]. Models were bivariate, that is, included two responses, ln ($\pi_{4D}$) and ln($\pi_{0D}$), which were assumed to covary and follow a Gaussian distribution. Only the 32 species with data for all seven predictors were included. Fixed effects were z-transformed when continuous so that estimated effect sizes were comparable for a given response. Phylogeny was included in the model as a random effect based on the inverse matrix of branch lengths in the maximum likelihood species phylogeny (Fig. 1). For the random effect and residual variance we assumed parameter expanded priors from a scaled F-distribution. A maximal model, containing all seven predictors as fixed effects, was constructed and then simplified by backwards elimination of predictors. The minimal model therefore only contains predictors with a significant ($\alpha \leq 0.05$) effect.

**Estimating genome size by flow cytometry.** To estimate the size of the genome for each species we followed the protocol outlined by ref. [66], with some minor modifications. In short, head tissue of butterflies (frozen fresh and preserved at −80 °C) were ground in Gailbraith's buffer and filtered through a 40-μm mesh, resulting in a suspension of free nuclei with minimal cell debris. The solution was centrifuged at 350/500 × g for 1 min, then the pellet of nuclei was resuspended in 300 μl propidium iodide (50 μg/ml; Sigma-Aldrich) and RNAse A (100 μg/ml; Sigma-Alrich) for staining and removal of RNA. After 1–2 h, fluorescence was measured using a BD LSR Fortessa running `Diva v8.0.1`. DNA content of cells were evaluated by propidium iodide binding using a 561 nm excitation laser and fluorescence emission at 600–630 nm. Each butterfly sample was measured alongside a sample of larval head plants (LHP) and HOST database[72]. Species were characterised as monophagous when LHPs were limited to one family or polyphagous when LHPs represented multiple families. Each butterfly sample was measured alongside a sample of *Drosophila melanogaster* (Oregon-R strain, genome size of ≈175 Mb[67]) to establish a reference genome position. Single nuclei were identified by plotting area versus width for the DNA labelling with 5–50 k positive nuclei recorded. For analysis, G0/1 peaks were gated for both the *D. melanogaster* and butterfly cells and relative intensities were then used to determine the genome size of the butterfly species using FlowJo v9.6.

**Life-history, karyotype and geographic range data.** Current census sizes were estimated as the product of geographic range and density. All species in this study can be found in the region of Catalonia, Spain, where butterfly monitoring has been taking place since 1994[68]. Density estimates were calculated as the mean number of individuals of each species seen per transect where that species is found, per year. The area range of each species was estimated from GBIF occurrence data (see Supplementary Data 2). The R package *rgbif*[69] was used to retrieve occurrence records—human observations with complete latitude and longitude information—for each species. Convex polygon areas (km²) were calculated using the function *eoo* in the R package *red*[70]. For species with large ranges, this was done separately for each land mass (to avoid including large bodies of water).

A list of larval host plants (LHP) for each species was compiled from ref. [71] and HOST database[72]. Species were characterised as monophagous when LHPs were limited to one family or polyphagous when LHPs represented multiple families. Mean forewing length (across at least ten individuals per sex) reported in ref. [73] was used as a proxy for adult body size. The mean between sexes was used for statistical analysis. Estimates of egg volume were retrieved from ref. [74], haploid chromosome number from ref. [73] and information on voltinism from ref. [71]. Since the number of generations can vary within species, we only classified species as monovoltine if they had strictly one generation per year throughout their European range and polyvoltine if otherwise. In species with variable chromosome numbers, the mean was used for statistical analyses. All data can be found in the Source Data file.

**Compliance with ethical standards.** Field sampling of butterflies was conducted in compliance with the School of Biological Sciences Ethics Committee at the University of Edinburgh and the ERC ethics review procedure. Permissions for field sampling were obtained from the Generalitat de Catalunya (SF/639), the Gobierno de Aragon (INAGA/500201/24/2018/0614 to Karl Wotton) and the Gobierno del Principado de Asturias (014252).

**Reporting summary.** Further information on research design is available in the Nature Research Reporting Summary linked to this article.

## Data availability

The source data used for MCMCglmm analyses in this study underlying Table 1, Figs. 1 and 2 and Supplementary Table 1 and Supplementary Figs. 2–8 are available in the

Source Data file. Raw reads are deposited at the European Nucleotide Archive under project PRJEB31360.

## Code availability

Python code for computing diversity, the MCMCglmm R code and the RAxML phylogeny are deposited at www.github.com/DRL/mackintosh2019. The *Mathematica* code used to generate Supplementary Fig. 8 is available as Supplementary Software.

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

## Acknowledgements

We thank Jarrod Hadfield for help with MCMCglmm and discussions throughout and for comments on an earlier version of this paper, Lisa Cooper for excellent work in the wetlab and Carla and Oskar Lohse for their enthusiastic support in the field. We are indebted to Cecilia Corbella, Michael Jowers, Karl Wotton, Luis Valledor, Martim Melo, Jose Campos and Megan Wallace for help with field and lab logistics, Paul Jay for contributing samples, Richard Lewington from Collins Butterfly Guide for permission to reproduce illustrations and Nicholas Galtier for sharing data and genetic diversity estimates. Permissions for field sampling were obtained from the Generalitat de Catalunya (SF/639), the Gobierno de Aragon (INAGA/500201/24/2018/0614 to Karl Wotton) and the Gobierno del Principado de Asturias (014252). This project was supported by an ERC starting grant (ModelGenomLand) and an Independent Research fellowship from the Natural Environmental Research Council (NERC) UK (NE/L011522/1) to K.L.. A.M. was supported by a summer studentship from the Institute of Evolutionary Biology at Edinburgh University, A.H. is supported by a Biotechnology and Biological Sciences Research Council David Phillips fellowship (BB/N020146/1) and R.V. is supported by project CGL2016-76322-P (AEI/FEDER, UE).

## Author contributions

K.L. and A.M. designed the study. A.H., K.L., A.M. and R.V. collected samples. A.M. and R.V. compiled the metadata. A.M. and M.W. performed laboratory work. A.M. and D.L. analysed the data with input from K.L. and B.C. A.M. and K.L. wrote the paper with input from all authors.

## Additional information

**Competing interests:** The authors declare no competing interests.

