## [Peer Review File · Nature Communications]

Reviewers' Comments:

Reviewer #1:

Remarks to the Author:

Summary

Mackintosh et al. present transcriptome-based estimates of polymorphism for many related species of butterflies. They use these data to uncover the correlates of genetic diversity at both neutral and strongly selected sites. Their primary conclusion is that the number "of chromosomes (and presumably total map length), as well as the body size, is correlated with genetic diversity at both neutral and selected sites, but that many other possible factors are not. This study is ambitious and generally competently done and well written. However, it is trying to cover a lot of ground and the stated conclusions are not strongly supported.

Major Concerns

One central concern is that the authors have tried to cover a lot of ground--maybe more than is feasible given their dataset. For example, despite testing a sizeable number of possible correlates, no concern about multiple testing is considered. The two putatively correlated features provide only a fairly weak rejection of the null (and in the case of chromosome number, no rejection when two outliers are removed). As a consequence, it is very challenging to draw strong conclusions from this work and I am left wondering what precisely has been learned.

The major strength of this study is that each sampled species is relatively closely related and many of the potential correlates found in previous work (e.g. romiguier et al.) are less likely to affect the conclusions here. This could be emphasized more explicitly as it is otherwise challenging to tell why this study is necessary from the introduction. In fact, at points it seems like the authors are underselling this. E.g., "While butterflies exhibit comparatively little variation in life-history strategy, one may still expect fecundity traits (e.g. relative egg size and voltinism." I recommend a significant rephrase to emphasize this feature of the presented work.

It is strange that that authors spend so little of their discussion addressing the differences between this and previous comparative population genomic studies. For example, the fact that there is no correlation between propagule size and $4D\text{-}\pi$ when this was the primary factor explaining neutral diversity in previous work. Instead the authors spend a lot of time discussing how natural selection might contribute, with some very back of the envelope math. Since this is one of the primary conclusions they reach, more of the discussion should focus on the life history contributions and ideally some resolution between this and previous studies considered.

The discussion becomes fairly technical, particularly with regard to the models of selection. I am concerned that this might not be a sufficient interest for the general readership of Nature Communications. Additionally, the level of approximation and assumption, particularly in the absence of any useful recombination rate estimates, makes it quite challenging to interpret the discussion as presented. I strongly suspect that a wide range of both background selection, neutral, and positive selection models could be consistent with the data. Of course, the genomic context (local recombination rate, and functional density), would help tremendously with clarifying the meaning and bounding the possible range of parameters necessary to explain these data.

A causative relationship between recombination rate and mutation rate would also explain much of the data presented. While the evidence for this type of relationship has been mixed, it would be useful to consider the possibility. Furthermore, it would be fairly straightforward to use divergence data among

species that show recent changes in chromosome number could be used to exclude the possibility.

Butterflies are ZW and a female would have hemizygous copies of many genes. This has the potential to impact estimates of diversity, particularly if a chromosome fusion event has included a Z chromosome. Furthermore, as patterns of selection on hemizygous sex chromosomes often differ from autosomes, their inclusion has the potential to further bias results (e.g. for the DFE-based analysis).

Minor

Page 2. "diversity in natural populations generally remains well below this". Not a strong enough statement. I suggest a rephrase. Neutral polymorphism has never even approached 0.75 per site in any group studied.

Page 3. "For simplicity, we restricted the estimation of synonymous diversity to fourfold degenerate sites (π_{4D}), as these genic sites are the least constrained by selection and can be assumed to be nearly neutral." There are many works showing that 4D sites are not strictly neutral. These should be cited in the introduction and the possible impacts discussed later in the paper.

Page 3. "This is because the increase in diversity due to reduced genetic drift is counteracted by the removal of diversity due to more efficient selection. We would therefore expect a weaker correlation for sites that are directly affected by selection than for neutral, linked sites. In contrast, any trait that affects non-synonymous genetic diversity (π_{0D}) via the absolute strength of selection (i.e. affects s but not N_e) should be more strongly correlated with diversity at non-synonymous sites (π_{0D}) than synonymous sites (π_{4D}), which are only indirectly affected." This section lays out some interesting, albeit simplified, ways of thinking about these data. However, these ideas are not reference again and instead the authors apply standard nearly neutral theory when thinking about the relationship of 4D and 0D sites. So, I am not clear about what this section of the introduction is meant to accomplish.

Reviewer #2:

Remarks to the Author:

This well-written, rich manuscript reports on a comparative analysis of the genetic diversity across 38 species of European butterflies. It is found that (i) genetic diversity at four-fold degenerate codon positions (π_{4D}) varies by an order of magnitude among species, (ii) π_{4D} is not correlated with species range, density or number of host plants, (iii) π_{4D} is significantly related to adult body size, but not egg size, (iv) π_{4D} is positively correlated to chromosome number, but not to genome size. The results are interpreted in the context of the long-term discussion, known as Lewontin's paradox, on the seemingly too narrow range of variation in genetic diversity across species. Of note, in addition to RNAseq data, genome size data were here newly generated in 37 species. Besides, two additional results not directly related to Lewontin's paradox are reported: (v) π_{4D} is not correlated to mitochondrial diversity, and (vi) the comparison of π_{4D} to π_{0D} , the genetic diversity at 0-fold degenerated codon positions, suggests a substantial contribution of mildly deleterious mutations (Gamma model for the DFE, estimated shape parameter=0.44).

There are, in my opinion, many interesting aspects in this paper. Recent advances on Lewontin's paradox (Leffler et al 2012, Romiguier et al 2014, Corbett-Detig et al 2015, Chen et al. 2017) were based on comparisons between distantly related species across animals and plants. One major strength of this ms is to produce a comprehensive (38 species) analysis at a much smaller taxonomic scale, across species sharing similar body plan, life history, ecology and, to some extent, recent

population history (all samples are from the Iberian refugium). This strength could be even more emphasized.

This is why it is important to know that the main findings of Romiguier et al (2014), i.e., an effect of life history traits but no effect of geographic range/current abundance/ecology on π_4D , are here corroborated. One comment, though: a quick re-analysis of the supplementary data shows that adult size and egg size are significantly correlated in this data set, and that the correlation coefficient between π_4D and egg size is associated with a p-value <0.1 . This might be discussed, and the results are perhaps even closer to Romiguier's than the manuscript currently suggests.

The other major result of the ms is the report of a significant relationship between π_4D and chromosome number, nc . This is superficially consistent with the suggestion that linked selection, which is expected to be more effective in large than small N_e , could be an important driver of the among-species variation in genetic diversity (Corbett-Detig et al 2015). Coop (2016), however, recalled that the Corbett-Detig effect, which is of the order of a factor of 2, is too small to solve Lewontin's paradox, and this is predicted by the population genetic theory. Building on Coop's work, the current ms reaches a similar conclusion, i.e., that polymorphism removal by linked selection is unlikely to explain the nc effect in butterflies, unless sweep rate is unplausibly high.

So the authors suggest that the nc effect could be explained by karyotypes evolving towards extreme chromosome numbers in periods of low N_e in butterflies. Various case studies consistent with this hypothesis are mentioned. None of these, however, are demonstrative, and the proposed scenario lacks any idea of why $nc=31$ would be the optimal state in butterflies. The ms correctly recalls the *Leptidea* example, in which low genetic diversity is associated with elevated nc (ref 53 and 54), consistently with the authors's hypothesis, but fails to mention that in *Leptidea* this change is accompanied by a rapid increase in genome size - which does not fit the broad pattern reported in this ms. Knowing that the relationship between nc and π_4D is not so strong and largely driven by two data points, I feel like this aspect of the ms occupies too much space, and does not deliver a 100% clear message.

Reviewers' comments:

Reviewer #1 (Remarks to the Author):

Summary

Mackintosh et al. present transcriptome-based estimates of polymorphism for many related species of butterflies. They use these data to uncover the correlates of genetic diversity at both neutral and strongly selected sites. Their primary conclusion is that the number of chromosomes (and presumably total map length), as well as the body size, is correlated with genetic diversity at both neutral and selected sites, but that many other possible factors are not. This study is ambitious and generally competently done and well

written. However, it is trying to cover a lot of ground and the stated conclusions are not strongly supported.

We thank the reviewer for the constructive and overall positive comments.

Major Concerns

One central concern is that the authors have tried to cover a lot of ground--maybe more than is feasible given their dataset. For example, despite testing a sizeable number of possible correlates, no concern about multiple testing is considered. The two putatively correlated features provide only a fairly weak rejection of the null (and in the case of chromosome number, no rejection when two outliers are removed). As a consequence, it is very challenging to draw strong conclusions from this work and I am left wondering what precisely has been learned.

We agree that the size of our dataset (like that of all comparative datasets that have been analysed in this context) imposes hard limits on the number of correlates that can meaningfully be explored. Because of this, we have from the outset restricted our analysis to seven potential correlates for which clear prior expectations (based on theory and/or previous comparative analyses) exist. While we agree with the reviewer's assessment that exploring the relations between even such a small number of predictors is ambitious, we believe that our main findings are both robust and interesting. We find that genetic diversity:

1. does not correlate with estimates of census size (in line with previous analyses across larger taxonomic scales)
2. does not correlate with propagule size (in contrast to the findings of Romiguier et al 2014)
3. correlates -vely with body size (found by several previous comparative studies)
4. correlates +vely with chromosome number (a new result)

We emphasize that our analysis which explores all potential correlates in a single multivariate model (and the fact that we present both the results of the full and the simplified model) goes beyond previous comparative studies which conduct and report separate tests for predictors. While we are not aware of any comparative study of genetic diversity that controls for multiple testing, this is of course a valid concern. We have explored this and have added text to the revised version (Results) to clarify that the body size effect (but not the chromosome number correlation) would be robust to Bonferroni correction (which is overly conservative).

The major strength of this study is that each sampled species is relatively closely related and many of the potential correlates found in previous work (e.g. romiguier et al.) are less likely to affect the conclusions here. This could be emphasized more explicitly as it is otherwise challenging to tell why this study is necessary from the introduction. In fact, at points it seems like the authors are underselling this. E.g., "While butterflies exhibit comparatively little variation in life-history strategy, one may still expect fecundity traits

(e.g. relative egg size and voltinism." I recommend a significant rephrase to emphasize this feature of the presented work.

Many thanks for this suggestion! The motivation for focusing our comparative analysis on butterflies was indeed to assess to what extent correlates that have been identified across the tree of life can explain the substantial variation in genetic diversity seen within major groups of animals. The high quality of the metadata available for butterflies (especially on geographic range, abundance and host plant range) allows us to not only better explore potential correlates of genetic diversity for which strong prior predictions exist, but also investigate the effect of ecological traits (larval host plant range) that have not been studied previously. We agree that both the strength of our design and choice of model group were not made clear enough in the previous version. We have re-written both the introduction and discussion (with a new subsection) to set this out much more clearly.

It is strange that that authors spend so little of their discussion addressing the differences between this and previous comparative population genomic studies. For example, the fact that there is no correlation between propagule size and $4D\text{-}\pi$ when this was the primary factor explaining neutral diversity in previous work.

Instead the authors spend a lot of time discussing how natural selection might contribute, with some very back of the envelope math. Since this is one of the primary conclusions they reach, more of the discussion should focus on the life history contributions and ideally some resolution between this and previous studies considered.

We agree that too little space was devoted to discussing the absence of any correlation between propagule size and $4D\text{-}\pi$. We have rewritten the first paragraph of the discussion to address how our findings can be resolved with comparative studies across larger phylogenetic scales.

The discussion becomes fairly technical, particularly with regard to the models of selection. I am concerned that this might not be a sufficient interest for the general readership of Nature Communications. Additionally, the level of approximation and assumption, particularly in the absence of any useful recombination rate estimates, makes it quite challenging to interpret the discussion as presented. I strongly suspect that a wide range of both background selection, neutral, and positive selection models could be consistent with the data.

Our motivation for discussing the analytic predictions for the effect of selection on neutral diversity was to provide a quantitative sanity check (albeit necessarily an approximate one) on the plausibility for the effect of chromosome number on diversity we find. Contrary to the suggestion that "a wide range of both background selection, neutral, and positive selection models could be consistent with the data" our calculations shows that: i) while BGS is irrelevant, ii) strong positive selection, i.e. a high rate of sweeps is required to explain the observed slope of $\ln(\pi_{4D})$ on chromosome number. We agree with the reviewer that these arguments have to be made in a way that is accessible to a general readership and have made every effort to strike a balance between being succinct (e.g. relegating derivations to S4 and some of the analysis to S5) whilst being clear about the assumptions and approximations involved. We strongly feel that this section is not only important for interpreting our results but also contains a novel and general result: the fact that one expects a linear relationship between the logarithm of neutral genetic diversity and chromosome number with a maximum slope $1/n_c$ while clear in hindsight only

became apparent to us once we have done the maths. It will be interesting to test this prediction more generally and in a wide range of taxa (which should be very feasible given the detailed information on chromosome numbers available for many groups). Following this comment, we have condensed this section of the Discussion further.

Of course, the genomic context (local recombination rate, and functional density), would help tremendously with clarifying the meaning and bounding the possible range of parameters necessary to explain these data.

Completely agreed. However, apart from the data compiled by Corbett-Detig (which are heavily biased towards lab models and domesticated species), we are not aware of any comparative datasets for which this level of genomic information is available, so this is clearly beyond the scope of our study. We very much hope that our work will motivate researchers to generate such data (especially dense recombination maps) that would allow to better model the population level forces that shape the distribution of genetic diversity both between species and along the genome. We have rewritten the Discussion to emphasize this point.

A causative relationship between recombination rate and mutation rate would also explain much of the data presented. While the evidence for this type of relationship has been mixed, it would be useful to consider the possibility. Furthermore, it would be fairly straightforward to use divergence data among species that show recent changes in chromosome number could be used to exclude the possibility.

We agree that a causative relationship between mutation and recombination could indeed partially explain the correlation between chromosome number and genetic diversity we find. We have added text to discuss this possibility (and the fact that at least from studies in *Drosophila* there is no strong evidence for it). We also point out that if recombination was mutagenic, we would expect the two species with strongly reduced chromosome number to have disproportionately low π_{4D} (given their ratio of π_{0D} and π_{4D}), i.e. we would expect them to fall below the line of best fit in Figure 2, which is not the case.

Butterflies are ZW and a female would have hemizygous copies of many genes. This has the potential to impact estimates of diversity, particularly if a chromosome fusion event has included a Z chromosome. Furthermore, as patterns of selection on hemizygous sex chromosomes often differ from autosomes, their inclusion has the potential to further bias results (e.g. for the DFE-based analysis).

This is an excellent suggestion. We completely agree that the inclusion of Z linked genes in our estimates of diversity is potentially problematic and thank the reviewer for raising this issue. To assess the impact of Z-linked loci we have identified and removed putative Z-linked contigs. in the 27 species for which we have both female and male samples, these would be expected to be homozygous in all female individuals. For each of the 1,314 conserved single copy orthologue (SCO) clusters, we tallied the number of species where the female sample is homozygous (HOM_F), the male sample is homozygous (HOM_M), and both samples are homozygous (HOM_both) (across all sites, rather than just 0D and 4D sites). The number of species in each of the three categories as a proportion of total species present in each cluster is shown in Figure S9. None of the 1,314 conserved SCO clusters contains transcripts that are homozygous in all females. However, since we do not know how conserved Z-chromosomes are across the analysed species, this is not sufficient to rule out putative Z-linkage. The distribution in Figure S9 reveals 37 SCO clusters which

display a high proportion of HOM_F species ($\geq 50\%$) with lower proportion of HOM_M and HOM_both species. We have redone all analyses without these contigs and find that this has almost no impact on our estimates of 0D and 4D (Figure S10). We have updated the Methods & all results to explain and incorporate this filter.

Minor

Page 2. "diversity in natural populations generally remains well below this". Not a strong enough statement. I suggest a rephrase. Neutral polymorphism has never even approached 0.75 per site in any group studied.

Rephrased as suggested.

Page 3. "For simplicity, we restricted the estimation of synonymous diversity to fourfold degenerate sites (π_{4D}), as these genic sites are the least constrained by selection and can be assumed to be nearly neutral." There are many works showing that 4D sites are not strictly neutral. These should be cited in the introduction and the possible impacts discussed later in the paper.

We have included references to studies showing selection in the form of codon usage bias at 4D sites and discuss the likely impact of our choice of filters SCO on our estimates of diversity at 0D and 4D sites.

Page 3. "This is because the increase in diversity due to reduced genetic drift is counteracted by the removal of diversity due to more efficient selection. We would therefore expect a weaker correlation for sites that are directly affected by selection than for neutral, linked sites. In contrast, any trait that affects non-synonymous genetic diversity (π_{0D}) via the absolute strength of selection (i.e. affects s but not N_e) should be more strongly correlated with diversity at non-synonymous sites (π_{0D}) than synonymous sites (π_{4D}), which are only indirectly affected." This section lays out some interesting, albeit simplified, ways of thinking about these data. However, these ideas are not reference again and instead the authors apply standard nearly neutral theory when thinking about the relationship of 4D and 0D sites. So, I am not clear about what this section of the introduction is meant to accomplish.

We feel that this section is important because it sets out alternative predictions for how traits affecting either N_e or s should correlate with π_{0D} and π_{4D} . We have condensed this section to hopefully clarify this point.

Reviewer #2 (Remarks to the Author):

This well-written, rich manuscript reports on a comparative analysis of the genetic diversity across 38 species of European butterflies. It is found that (i) genetic diversity at four-fold degenerate codon positions (π_{4D}) varies by an order of magnitude among species, (ii) π_{4D} is not correlated with species range, density or number of host plants, (iii) π_{4D} is significantly related to adult body size, but not egg size, (iv) π_{4D} is positively correlated to chromosome number, but not to genome size. The results are interpreted in the context of the long-term discussion, known as Lewontin's paradox, on the seemingly too narrow range of variation in genetic diversity across species. Of note, in addition to RNAseq data, genome size data were here newly generated in 37 species. Besides, two additional results not directly related to Lewontin's paradox are reported: (v) π_{4D} is not

correlated to mitochondrial diversity, and (vi) the comparison of π_{4D} to π_{0D} , the genetic diversity at 0-fold degenerated codon positions, suggests a substantial contribution of mildly deleterious mutations (Gamma model for the DFE, estimated shape parameter=0.44).

We are grateful for this enthusiastic summary of our work!

There are, in my opinion, many interesting aspects in this paper. Recent advances on Lewontin's paradox (Leffler et al 2012, Romiguier et al 2014, Corbett-Detig et al 2015, Chen et al. 2017) were based on comparisons between distantly related species across animals and plants. One major strength of this ms is to produce a comprehensive (38 species) analysis at a much smaller taxonomic scale, across species sharing similar body plan, life history, ecology and, to some extent, recent population history (all samples are from the Iberian refugium). This strength could be even more emphasized.

We agree that this was not sufficiently emphasized and, following this suggestion which was made by both reviewers, have rewritten the relevant sections of the introduction and discussion (see our responses to reviewer 1).

This is why it is important to know that the main findings of Romiguier et al (2014), i.e., an effect of life history traits but no effect of geographic range/current abundance/ecology on π_{4D} , are here corroborated. One comment, though: a quick re-analysis of the supplementary data shows that adult size and egg size are significantly correlated in this data set, and that the correlation coefficient between π_{4D} and egg size is associated with a p-value <0.1 . This might be discussed, and the results are perhaps even closer to Romiguier's than the manuscript currently suggests.

This is a very helpful comment. While Romiguier et al (2014) find a stronger correlation between π_{4D} and (absolute) propagule size than body size across animals, we see the opposite pattern in butterflies: the correlation between $\log(\pi_{4D})$ and body size is strong $p < 0.010$, whereas the correlation between $\log(\pi_{4D})$ and (absolute) egg volume is weak and non-significant $p < 0.1$. If egg volume were a good predictor of π_{4D} independently of body size (i.e. explaining different variation), we should see a correlation between relative propagule size ('egg volume / body size') and genetic diversity; which is not the case. We have rewritten the relevant parts of the discussion to make this much clearer.

The other major result of the ms is the report of a significant relationship between π_{4D} and chromosome number, n_c . This is superficially consistent with the suggestion that linked selection, which is expected to be more effective in large than small N_e , could be an important driver of the among-species variation in genetic diversity (Corbett-Detig et al 2015).

Of course our discussion goes further than showing that this finding is superficially consistent with the action of linked selection: we quantify the frequency of positive selection that would be required to generate the observed correlation.

Coop (2016), however, recalled that the Corbett-Detig effect, which is of the order of a factor of 2, is too small to solve Lewontin's paradox, and this is predicted by the population genetic theory. Building on Coop's work, the current ms reaches a similar

conclusion, i.e., that polymorphism removal by linked selection is unlikely to explain the n_c effect in butterflies, unless sweep rate is unplausibly high.

The rate of sweeps that would be required to explain the correlation is indeed high. We also find that linked selection due to sweeps alone gives a rather poor fit to the range of diversity we see across species of butterfly (Fig. S6) and so can only be part of the answer.

So the authors suggest that the n_c effect could be explained by karyotypes evolving towards extreme chromosome numbers in periods of low N_e in butterflies. Various case studies consistent with this hypothesis are mentioned. None of these, however, are demonstrative, and the proposed scenario lacks any idea of why $n_c=31$ would be the optimal state in butterflies.

This is a good point. We agree that the forces driving karyotype evolution in butterflies are currently not well understood and have reworded our argument to make it clear that this it relies on supposing that an optimum number of chromosomes.

The ms correctly recalls the *Leptidea* example, in which low genetic diversity is associated with elevated n_c (ref 53 and 54), consistently with the authors' hypothesis, but fails to mention that in *Leptidea* this change is accompanied by a rapid increase in genome size - which does not fit the broad pattern reported in this ms. Knowing that the relationship between n_c and π_{4D} is not so strong and largely driven by two data points, I feel like this aspect of the ms occupies too much space, and does not deliver a 100% clear message.

We agree that the increase in genome size seen in *Leptidea* is indeed compatible with Lynch's suggestion that genomes expand in drift-dominated populations and have clarified this in the discussion. However, in contrast to this observation ($N=1$), our results across species ($N=38$) shows that there is no evidence for a general relation between genetic diversity and genome size in butterflies. While we agree that the two species with lowest n_c have (unsurprisingly) a disproportionate effect on the estimated relationship between n_c and genetic diversity, we stress that a positive relationship between diversity and chromosome number remains when those two species are removed.

Reviewers' Comments:

Reviewer #1:

Remarks to the Author:

I am satisfied with the authors' responses and revised manuscript. It is a valuable contribution.

Reviewer #2:

Remarks to the Author:

I'm happy with the responses of the authors to my comments, which were mostly minor, and the associated revisions.